# Student Stress and Online Shopping Addiction Tendency among College Students in Guangdong Province, China: The Mediating Effect of the Social Support

**DOI:** 10.3390/ijerph20010176

**Published:** 2022-12-22

**Authors:** Huimin Li, Xinyue Ma, Jie Fang, Getian Liang, Rongsheng Lin, Weiyan Liao, Xuesong Yang

**Affiliations:** 1School of Human Resources, Guangdong University of Finance & Economics, Guangzhou 510320, China; 2The First Affiliated Hospital of Wenzhou Medical University, Wenzhou 325000, China; 3School of Psychology, South China Normal University, Guangzhou 510631, China; 4Division of Histology and Embryology, Key Laboratory for Regenerative Medicine of the Ministry of Education, Medical College, Jinan University, Guangzhou 510632, China

**Keywords:** social support, online shopping addiction tendency (OSAT), student stress, mediating effect, college students

## Abstract

Online shopping addiction tendency (OSAT) among college students has become too serious to ignore. As a result, it is necessary to carefully examine the relevant factors that shape students’ online shopping addiction tendencies. This study aimed to determine whether social support mediates the relationship between college students’ stress (academic hassle, personal hassle, and negative life events) and OSAT. In this cross-sectional study using a convenient sampling method, Chinese students from eight universities in Guangdong Province, China, completed self-administered questionnaires in either printed or online format. The survey data includes daily online shopping usage, college student stress, a social support rating scale, an online shopping addiction tendency scale for college students, and demographic information. A total number of verified and valid questionnaires were returned. In a sample of 1123 (mean age = 20.28 years; 58% females). Each individual had online shopping experience. The survey revealed no gender differences in OSAT. There was a statistically significant relationship between student stress (academic hassle, personal hassle, and negative life events) and students’ OSAT scores and social support. The latter was negatively correlated with OSAT and mediated the relationship between college students’ stress and their OSAT. In conclusion, university students’ stress (academic hassle, personal hassle, and negative life events) acts as a trigger for OSAT. A combination of a high stress level and a lack of social support increases the likelihood of developing OSAT. Social support has an effect on the OSAT of college students by relieving their stress; social support is a protective factor against the OSAT for college students.

## 1. Introduction

According to the 50th Statistical Report on China’s Internet Development, the number of online shoppers in China has reached 1000.51 million by September 2022. College students have been the driving force behind online shopping, but they are also particularly susceptible to OSAT [1]. According to *The Core Information and Interpretation of Health Education for Adolescents in China* (2018 edition), compulsive online shopping is a form of internet addiction. According to a study by Lee et al., offline impulse purchases and internet addiction are strongly correlated with online impulse purchases [2]. Thus, OSAT is a combination of online addiction and offline impulse shopping. A survey of 1183 postgraduates from three universities in the province of Guangdong, China, revealed that 16.7% of them suffer from online shopping overload [3]. As a form of electronic commerce, online shopping enables consumers to purchase goods quickly and affordably, which can provide college students with superior service. However, college students have also been significantly affected by online shopping. Some college students spent more time and money on online shopping, resulting in a substantial increase in their living expenses and even higher levels of debt like offline impulse shopping [4]. Moreover, long-term online use would negatively impact the academic performance of college students because it wastes an excessive amount of time and energy [5]. Long-term use of mobile phones and other internet-accessing devices may result in deteriorating health syndromes [6,7]. Rose et al. proposed that among the factors predicting online shopping addiction, the medium of online retail, including enjoyment, social anonymity, and cognitive overload, was the most important [8]. Paulus et al. demonstrated that certain social factors (such as negative experiences and parental influences) are risk factors for internet gaming addiction [9]. Daily stress is positively correlated with Facebook usage intensity and addictive tendencies, whereas offline social support may protect mental health [10]. Previous research has identified predictive factors (e.g., personality traits, environmental factors, cognitive overload, and characteristics of internet retailing) for online shopping addiction [1,8,11]. Little empirical research has investigated the relationship among students’ stress, social support, and OSAT in order to identify the factors that may help to explain this relationship. Thus, we utilized a sample of college students from the province of Guangdong to examine whether or not student stress is a significant predictor of college students’ OSAT and to determine whether or not social support mediates this relationship.

Internet-based problem shopping in China is regarded as an important research project for two reasons. First, the precautionary measure against the COVID-19 pandemic accelerated the growth of internet shopping in China. It appears that the proliferation of online shopping has had both negative and positive effects. The issue of college students’ OSAT scores has garnered considerable attention. Second, it is imperative to conduct in-depth research on addiction-related risk factors to provide scientific justification for the prevention and treatment of OSAT in college students.

### 1.1. OSAT and Student Stress

OSAT is defined as a pattern of excessive, compulsive, and problematic online shopping behavior that results in economic, social, and emotional problems [1,12]. Consequently, as mentioned previously, it is considered a distinct form of internet addiction closely related to offline purchase addiction. As an uncontrolled problematic buying behavior, offline addictive buying is characterized by a severe and frequent urge to make unnecessary purchases, occasionally exceeding realistic budgeting, i.e., spending too much time shopping, which results in negative outcomes (such as high levels of stress and financial problems) [13,14,15,16]. Numerous studies [17,18,19,20] have found a strong correlation between stress and substance addiction. Due to the rise in popularity of the Internet, researchers have focused on the connection between stress and internet addiction. Numerous studies have demonstrated that stress has a positive predictive effect on internet addiction [21,22,23]. The study by Li et al. revealed a strong correlation between stressful life events and online game addiction among college students [23,24].

Academic stress is associated with various forms of internet addiction, including online gaming and mobile phone addiction, among adolescents [24,25]. In contrast, Velezmoro et al. [26] discovered that stress was a significant predictor of internet pornography addiction. When students leave home to attend college and undergo the transition from the family to the university environment, they are likely to experience a novel type of stress related to their adaptation to the academic community.

In actuality, student stress resulting from the transition to college life is caused by a variety of stressors, including financial concerns, the stress of independence, academic performance, and interpersonal relationships [27]. Li et al. reported that Chinese college students’ stress consisted of three components: academic difficulty (e.g., learning efficiency, competition with classmates in academics, and examination pressure), personal difficulty (e.g., improper teaching/education, lack of social skills, lack of love, and lack of interpersonal communication skills), and a negative life event (e.g., failed exams and being accused in public [28]. The stress can induce negative emotions such as anxiety and depression. Based on the theory of stress-coping strategies [29], there are two ways that individuals manage stress. The first is problem-focused coping, and its purpose is to solve the problem by taking action (such as seeking social support) to alter the source of stress. The second is emotion-focused coping, which aims to alleviate or control emotional distress. In an emotion-focused coping strategy, some individuals will seek relief through escapism, which may be a temporary means of avoiding destructive effects. Obviously, this is not a good long-term option. The excessive use of shopping websites may be the result of a desire to escape reality as well as a quick distraction from discomfort and instant relief from negative emotions [30]. Due to their innate affinity for shopping, women are more likely to develop an online shopping addiction [8].

These studies indicate that college student stress relief is essential for OSAT prevention and reduction.

### 1.2. Social Support as a Mediator

This article refers to offline social support when mentioning social support. It is commonly believed that social support encompasses the following types of support: (1) instrumental support, i.e., material and/or behavioral assistance; (2) informational support and appraisal, i.e., providing information to aid in problem-solving; and (3) emotional support, i.e., focusing on wraparound care, love, and sympathy [31]. In other words, nonprofessionals in formal support groups and informal help relationships attempt to provide adequate material and spiritual assistance to those in need [32]. According to House et al., social support is the assistance provided by those with interpersonal relationships with the recipients, such as family members, friends, coworkers, or other significant individuals [33]. The purpose of social support is to empower and enable people to overcome emotional distress as well as to protect those with physical, mental, or financial issues [34,35]. The classic buffering model of social support suggests that social support in stressful situations can mitigate the negative effects of stress on mental health [36]. The main-effect model of social support also demonstrates that social support can serve as “mental health care” and “timely help” for an individual’s physical and mental health and can increase mental health and psychological well-being regardless of the situation. When individuals receive more social support, they may exhibit greater self-control and reduced impulsivity in response to stressful events in order to conform their behavior to social norms and maintain rational thought [35,37]. Evidently, the provision and availability of social support for college students is a crucial factor in reducing psychological distress, enhancing mental health, and enhancing subjective well-being [38]. Numerous studies suggest that offline social support can preserve mental health and reduce addictive behavior [9,10,17,21]. On the contrary, online social support may predict a tendency toward internet addiction. For instance, researchers found that online social support was positively associated with Facebook addiction. The positive experience of online social support might contribute to the desire to receive more interaction and a sense of identity, which, however, may probably increase the individual’s vulnerability to developing Facebook addiction [10,39]. According to the mediated model, the effects of stress on internet addiction are mediated by a series of processes [22,23]. Researchers have focused on three major categories of mediating variables: individual characteristics, environmental characteristics, and internet use variables.

Stress influences internet addiction not only through the mediation of individual variable characteristics but also through the mediation of environmental factors. A study found that stress influences online addiction through the mediation of social support, specifically greater stress. The lower a person’s social support, the greater his or her Internet addiction [40,41]. However, between student stress and OAST, relatively little research has been conducted.

### 1.3. The Present Study

The purpose of this study is to examine Chinese college students in relation to variables like social support, student stress, and online shopping addiction, as well as whether social support mediates the effect of college students’ stress (i.e., academichassle, personal hassle, and negative life events) on their OAST, which have not received sufficient scholarly attention to date. In particular, we proposed the following hypotheses: (1) compared to low-stress groups, high-stress groups exhibit higher OSAT and lower social support, and female students are more susceptible to OSAT; (2) different dimensions of students’ stress (academic hassle, personal hassle, and negative life events), social support, and college students’ OSAT are significantly correlated; and (3) social support mediates the relationship between college student stress and OSAT.

## 2. Methods

### 2.1. Participants

Initially, a total of 1230 questionnaires were distributed using the convenience sampling method to students in various majors and grade levels at several colleges and universities in Guangdong Province, China. The final sample consisted of 1123 college students, 471 (41.9%) of whom were male and 652 (58.0%) of whom were female. They were all between the ages of 17 and 23 (mean age = 20.28, SD = 1.14). A total of 393 (34.9%) were recovered by first-year students, 229 (20.3%) by sophomores, 325 (28.9%) by juniors, 150 (13.3%) by seniors, and 26 (2.3%) by graduates or above.

### 2.2. Procedures

The assessment was administered both online and offline. The offline assessment was administered in the classroom, and participants who were unable to complete it offline took the online assessment. The majority of the participants were selected at random. Prior to the survey, research assistants were instructed in standard operating procedures and the strict confidentiality of participant information. All participants were asked to complete the survey in accordance with the same instructions, and informed consent and the right to withdraw at any time were guaranteed. Because all participants were over 18, parental consent was not necessary. This research was conducted in accordance with the Declaration of Helsinki and with the approval of the Guangdong University of Finance and Economics Ethics Committee.

### 2.3. Measures

#### 2.3.1. College Student Online Shopping Addiction Tendency (OSAT) Scale

The OSAT [42] was developed by Lang Xu et al. to assess the propensity for online shopping addiction among college students. The scale consists of 22 items that assess the following four dimensions: dysfunction (7 items, such as “I go out less and less because I am always shopping on websites”), over-consumption (7 items, such as “I always spend a lot of time browsing shopping websites every day”), withdrawal reactions (5 items, such as “When I want to shop online but there is no Internet, I become very distressed”), and pleasure of online shopping (3 questions, such as “I was very happy when I opened the delivery”). Each question on the survey was evaluated using a five-point Likert scale. College students’ responses were based on their actual circumstances, and the higher their total score, the greater their online shopping addiction. In this study, the scale’s Cronbach’s alpha correlation coefficient was 0.958.

#### 2.3.2. College Student Stress Scale

The Chinese College Student Stress Scale (CCSSS) was developed by Li et al. [28] and consists of 30 items, primarily assessing personal hassle (16 items, for example, “The family’s economic condition is poor”), academic hassle (10 items, for example, “The overall academic performance is not ideal”), and negative life events (4 items, e.g., “To be publicly accused”). The higher the score, the greater the pressure. This scale’s total score ranges from 0 to 90. In theory, 45 is used as a cutoff point, with pressures greater than 45 being considered high and those lower than 45 being considered low. In this study, the Cronbach’s alpha correlation coefficient for the overall scale was 0.967, and Cronbach’s alpha correlation coefficients for each subscale were 0.924, 0.923, and 0.821, respectively.

#### 2.3.3. Social Support Scale

Xiao developed the Chinese Social Support Rating Scale (CSSRS) based on China’s distinctive social and cultural environment [43]. It consists of 10 items. It is used to assess objective support (3 items, e.g., “What have been the sources of comfort and care you have received in times of distress in the past?”), subjective support (4 items, e.g., “How many close friends can you rely on for support and assistance?”) and utilization of social support (3 items, e.g., “What are the ways to help you when you are in trouble?”). Each item was scored on a four-point Likert scale, with a higher score. As the participants were all college students, some items were modified accordingly, such as the category “children” being removed, “colleagues” being replaced with “classmates”, “husband and wife” being replaced with “partners”, and “institution” being replaced with “college”. The modified scale had good internal consistency (Cronbach’s alpha correlation coefficient = 0.704).

### 2.4. Statistical Analysis

We utilized SPSS 21.0 to analyze the data. These steps were taken: Firstly, a descriptive analysis of all variables, demographic parameters of participants, and Pearson correlation coefficients were described. Secondly, we used Model 4 of the PROCESS macro for SPSS to test the mediating effects of social support [44]. Thirdly, the bootstrap confidence intervals (CIs) determine whether the effects in Model 4 are significant, based on 5000 random samples. An effect is regarded as significant if the CIs do not include zero. All study variables were standardized in Model 4 before data analyses. The level of statistical significance for the entire analysis in this study was set at 0.05.

## 3. Results

### 3.1. Analysis of Gender Difference

It is commonly believed that women are more likely to develop an online shopping addiction. Therefore, we used an independent sample t-test to investigate whether there is a gender difference in the online shopping addiction tendencies of Chinese college students. The results of the independent sample t-test indicated that there was no statistically significant difference between the online shopping addiction tendencies of male and female Chinese college students (*t* = 0.29, NS) (Table 1).

### 3.2. Analysis of the Difference in Levels of Stress

The top 30% of the stress scale score was designated as the high-stress group, while the bottom 30% was designated as the low-stress group. A t-test on independent samples was used to examine the differences in social support and online shopping addiction between college students in the high-stress group and those in the low-stress group.

There were significant differences in social support between college students in different stress groups (*t* = 7.579, *p* < 0.001), with students in the high-stress group reporting significantly less social support than those in the low-stress group. In addition, there were significant differences in the OSAT scores of college students in the various stress groups (*t* = −5.006, *p* < 0.001), with the high-pressure group having significantly higher OSAT scores than the low-pressure group (Table 2).

### 3.3. Descriptive Statistics and Correlations

Table 3 shows the mean, standard deviation, and Pearson correlation coefficients between stress, social support, and OSAT. As expected, personal and academic hassles and negative life events were positively correlated with OSAT and negatively correlated with social support. Total stress was also positively correlated with OSAT and negatively correlated with social support. Social support was negatively correlated with OSAT.

### 3.4. Mediation Model Analysis

When examined whether OSAT was derived from the total effect of stress, we found that the regression coefficient between college students’ stress and an online shopping addiction tendency was significantly different (*β* = 0.353, *p* < 0.001). It suggests that college students’ stress could positively predict their online shopping addiction tendency when only considering their stress in relation to their online shopping addiction tendency.

Figure 1 shows the mediation model we constructed, we used Model 4 of the PROCESS macro to test the mediating role of social support on the relationship between stress and online shopping addiction tendency (OSAT). After controlling for gender and grade, stress was negatively associated with social support (*β* = −0.368, *p* < 0.001), which in turn was negatively related to OSAT (*β* = −0.256, *p* < 0.001). The positive direct association between stress and OSAT remain significant (*β* = 0.258, *p* < 0.001). Therefore, social support partially mediated the relationship between stress and OSAT (indirect effect = 0.094, SE = 0.014, 95% CI = [0.062, 0.118]). The mediation effect accounts for 26.6% of the total effect of stress and OSAT (Table 4).

These results are sufficient for us to conclude that social support has a significant mediating effect on the relationship between college students’ stress and their tendency to engage in online shopping.

## 4. Discussion

Although there have been many studies on substance addiction, behavioral addiction, and social media addiction in the past, more research is needed. The distinguishing feature of online shopping addiction is that it shares characteristics with social media addiction and offline shopping addiction but is distinct from the aforementioned addictions.

There is no reason for online retailers to accept returns for seven days, keep track of customer preferences and purchase histories, send special information, let customers use computers and mobile devices to make purchases whenever they want, and use rich user interfaces, discount advertising data, and information from previous customers’ online product reviews, all of which will appeal to the general public and cash-strapped students, thereby encouraging online impulse buying.

This study aimed to determine if manipulation of student stress (academic hassle, personal hassle, and adverse life events) could accurately predict online shopping addiction in college students, if college students with high stress have lower social support and higher OAST, and if social support can mediate the relationship between them.

In contrast to the common belief that females are more likely to go shopping [45], we found that gender had no significant effect on the three variables (social support, students’ OSAT, and students’ stress) of our study. According to the traditional social division of labor [46], women are still primarily responsible for caring for children and housework, whereas men are primarily responsible for earning money to support their families. University life is distinct from family life; it is a unique environment. Everyone resides in a communal dormitory and is responsible for their own studies and lives; during the COVID-19 epidemic, the majority of students spent their days indoors. Students take classes online and purchase all supplies, including three meals per day, online. If they desire, they can also purchase meal delivery services online. This is a unique time, and it cannot be compared to the time before the pandemic. In actuality, scientific research frequently leads to a conclusion that contradicts common knowledge or presumptions. For instance, contrary to popular belief, studies have shown that smartphone addiction is unrelated to the gender of the user [47]. On the other hand, this also reminds us that male college students are the most likely subject that can be ignored, such as their propensity for online shopping addiction.

Comparing the higher-stress group to the lower-stress group, this study revealed that the higher-stress group had lower social support and a higher OSAT. This is generally consistent with similar research conducted in other parts of the world [48,49]. Social support is considered a protective factor, and it can predict the occurrence of online shopping addiction, which has a significant impact on college students’ OSAT. Thus, a high level of social stability support can alleviate the stress of college students. According to the buffering model of social support, social support as a stress buffer could alleviate the pressure on individuals and protect them from the detrimental effects of stressful life events. It can contribute in two ways. First, we can intervene in stressful situations by reducing or preventing individual stress assessment responses. In other words, the perception that others can and will provide the required resources may redefine the potential harm caused by stress, preventing it from being overestimated. Second, by providing solutions to problems, it can directly influence physiological processes, calm the endocrine system, and diminish the perception of pressure [50]. College students are also at a pivotal point in their lives as they are about to become new members of society. Those who perform less well in their studies or have difficulty advancing their careers are more likely to channel and alleviate their disappointment through online shopping, particularly when they perceive a lack of social support, which is consistent with previous research [8,30]. This is likely due to the fact that shopping creates a distorted alternative for social support and emotional rewards and satisfaction that cannot be obtained through studies or a career.

We found that the three categories of student stress (i.e., academic hassle, personal hassle, and negative life events) were negatively correlated with social support and positively correlated with OSAT. According to the ACE model of Internet addiction proposed by Yong [51], letters A, C, and E refer to anonymity, convenience (convention), and escape from reality based on the individual’s understanding and the network’s development. Anonymity allows individuals to disguise themselves and conceal their true identity. Convenience implies that the network brings people closer together in terms of communication, that there is no distance between them, and that its simple operation provides users with convenience. The term “escape” refers to the individual’s encounter with obstacles in the real environment, i.e., you can find solace in virtual space and escape from real problems. This theory has widespread scholarly support. Yee believed that the motivation to escape could accurately predict internet game addiction [52]. When college students are exposed to academic stress, personal stress (e.g., anxiety, worry, and discomfort), and adverse life events, some of them may choose to escape the real world through online shopping. Rich and colorful online shopping pages, as well as video interaction and timely communication, can make them feel delighted and amused during online social interaction. Payment for online shopping is quick and simple, and express delivery is available. Online shopping allows individuals to temporarily forget the pressures of reality, engage in a process of enjoyable seeking, and unconsciously engage in excessive consumption. Consistent with the findings of Cheung C M K et al. [53], this results in double losses of time and property, leading to online shopping dependency. This suggests that if we are to reduce OSAT among Chinese students, we must tend to them more carefully and take corrective action to alleviate their stress. Although the efficacy of this approach has been confirmed by other studies [26,54,55,56], the effect of student stress on social support alleviating online shopping addiction is still unknown.

Individual and environmental factors may interact, according to Banguela’s (1997) social cognitive theory, making the process of pressure affecting internet addiction more complex. We assume that social support mediates the relationship between student stress and OSAT scores. This hypothesis was determined by our research, and the results demonstrated that social support plays a role in reducing OSAT by relieving stress, which is generally consistent with previous research [10,49]. Social support processes operate through microscopic biological, psychological, or behavioral mechanisms that produce main effects, buffer effects, or both [57]. Results of social support make individuals feel loved, accepted, and valued, which meets people’s natural needs [58]. This suggests that stress, (e.g., academic stress, individual distress, and negative life events) can exacerbate the tendency of students to seek solace in online shopping to compensate for the lack of social support. Therefore, increasing the availability of social support is essential for assisting students in overcoming OSAT. Knowing that entering college for the first time signifies the beginning of an independent life as an adult can be stressful for most students. Certainly, supportive interpersonal relationships can assist students in coping with this crucial transition [59]. Previous research has demonstrated that social support can alleviate some of the burden of acculturative stress and improve academic performance as well as mental health symptoms [60,61]. Therefore, we hypothesized that social support would have a mediating effect on the reduction of students’ OSAT by buffering student stress; thus, the relevant hypothesis has been confirmed by this study. Consequently, social support is likely to have a protective effect on students from OSAT, either directly or indirectly via the mediating effect [62].

University educators should promptly identify college students experiencing stressful events and recognize that if they do not promptly provide them with psychological assistance, this issue will result in serious problems for the students.

It is essential that OSAT be managed in three different ways. The first is to develop software that displays a warning message when online purchases exceed a predetermined threshold. Secondly, the introduction of self-assessment questions to social networking sites assists users in determining their online purchasing status. Thirdly, provide students with specialized group and individual psychological counseling to assist them in navigating a difficult life stage.

## 5. Conclusions

Based on the foregoing analysis, this study reaches the following conclusions: First, we found no statistically significant gender differences on the measured dimensions of OSAT among Chinese college students in Guangdong. We demonstrated for the first time that the greater the pressure college students experienced, the greater the markers for additive OSAT and the lower their social support. Secondly, student stress (academic hassle, personal hassle, and negative life events) is positively correlated with the OSAT and negatively correlated with social support. Student stress is an indicator of online shopping addiction. Lastly, social support has a mediating effect on the relationship between student stress and OSAT. The significance of our research is to fill the gap in this field for the first time. Giving college students positive offline social support can alleviate their stress and reduce their tendency to become addicted to online shopping.

## 6. Limitations and Future Directions

One limitation is the potential applicability of this study and its findings beyond its strictly defined research subjects, i.e., college students, to the general population or other social groups. A second limitation is that we did not conduct intervention research. Therefore, the experimental basis for implementing effective intervention measures for controlling online shopping addiction could not be determined in advance through this study. In addition, the sample consisted of non-clinical participants, so college students’ online shopping addiction did not reach clinically elevated levels. Therefore, additional research is required to confirm the effect of social support and student stress in clinical samples with higher levels of students’ online shopping addiction. Thirdly, our cross-sectional study limits causal inferences to some degree. Authorities have not yet classified college students’ internet shopping addiction as a mental disorder [63], and the precise definition remains unclear. In addition, there is no international standard scale, so we can only use the measurement tools created by scholars (which have not been widely adopted) [12,42,64]. Future research will therefore concentrate on the associations among online social support, financial risk perception, academic procrastination, time perspective, self-esteem, coping style, and the OSAT. 

## Figures and Tables

**Figure 1 ijerph-20-00176-f001:**
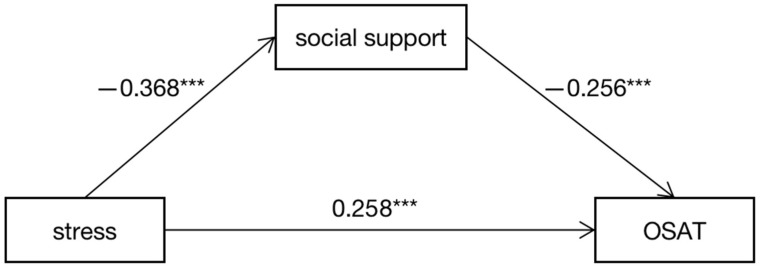
The mediation model: OSAT = Online Shopping Addiction Tendency; *** *p* < 0.001.

**Table 1 ijerph-20-00176-t001:** Characteristics of the sample (*N* = 1123, *M* ± *SD*).

		Student Stress	OSAT	Social Support
**Gender**	Male	65.32 ± 17.56	51.13 ± 15.21	14.55 ± 2.85
	Female	64.48 ± 16.54	50.69 ± 15.11	14.63 ± 2.96
** *t* **		0.497	0.292	−0.271

Note: OSAT = Online Shopping Addiction Tendency.

**Table 2 ijerph-20-00176-t002:** Characteristics of the sample (*N* = 1123, *M* ± *SD*).

	Social Support	OSAT
The high-stress group	13.52 ± 2.77	52.27 ± 13.88
The low-stress group	16.08 ± 2.34	45.40 ± 3.81
*t*	7.579 ***	−5.01 ***

Note: OSAT = Online Shopping Addiction Tendency; *** *p* < 0.001.

**Table 3 ijerph-20-00176-t003:** Cronbach’s alpha correlation coefficients, means, standard deviations, and Pearson’s correlation coefficients between the study variables.

		*M*	*SD*	1	2	3	4	5	6
1	Personal Hassle	29.84	10.42	*0.924*					
2	Academic Hassle	26.81	9.62	0.923 ***	*0.923*				
3	Negative Life Events	11.49	4.03	0.902 ***	0.890 ***	*0.821*			
4	Total Stress	64.95	17.11	0.738 ***	0.737 ***	0.690 ***	*0.967*		
5	OSAT	50.94	15.15	0.166 ***	0.173 ***	0.173 ***	0.353 ***	*0.958*	
6	Social Support	14.58	2.89	−0.216 ***	−0.203 ***	−0.171 ***	−0.368 ***	−0.351 ***	*0.704*

Note: *N* = 1123; Cronbach’s alpha correlation coefficients are in diagonal bold italics; *** *p* < 0.001.

**Table 4 ijerph-20-00176-t004:** Parameter estimates of the study model.

Effect	*β*	SE	95% CI	
			Lower	Upper
Direct effects				
Total Stress→Social Support	−0.368 ***	0.034	−0.435	−0.302
Total Stress→OSAT	0.258 ***	0.063	0.129	0.380
Social Support→OSAT	−0.256 ***	0.030	−0.317	−0.200
Indirect effects				
Total Stress→Social Support→OSAT	0.094 ***	0.014	0.062	0.118

Note: N = 1123; OSAT = Online Shopping Addiction Tendency; *** *p* < 0.001.

## Data Availability

The data presented in this study are available on request from the frist authors (H.L.).

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
