# Peer review of "Student Stress and Online Shopping Addiction Tendency among College Students in Guangdong Province, China: The Mediating Effect of the Social Support"

_ijerph, 2022, doi:10.3390/ijerph20010176_

Round 1
Reviewer 1 Report
It has been a pleasure to review this paper that analyses the relationship between stress among Chinese university students and online shopping addiction. The rise of this type of addiction requires education and awareness campaigns, so I consider it an interesting article for the journal. The introduction is well structured and provides literature to contextualise the study. The method is clearly explained. The results are adequately represented. There is a good discussion of the results, although I would miss a more forceful recommendation or proposal for society, given that the study detected the problem and the research is there to solve it (Lines 366-368 and 380-382). In lines 417-420, they consider that university educators are the ones who should discover this problem in time, how? when should psychological help be given? Finally, the article ends with a clear conclusion. Addiction to online shopping requires a certain economic level, perhaps it would be interesting for future research to include the economic aspect as another variable. Congratulations on the study.
Reviewer 2 Report
There are several factors which might contribute to OSAT which could be mentioned:
· the hyperconnectivity of Gen Y and Gen Z to different devices (e.g., phones, laptops, watches) sets people for messaging through conventional and social media, opportunity, and impulsive buying; the messaging is often customized so that the products are compelling based on prior shopping history and preferences
· the liberal return policies contribute to impulsive shopping behaviors as the decisions are not as final
· the convenience of ordering 24/7 and the lures of (free) and often fast delivery are also compelling motivators for OSAT
I would like to see some mention of how the findings may also generalize to the general adult population instead of the implication that the phenomena discussed were college student specific
Reviewer 3 Report
1. I don't think the study cannot be easily extrapolated to all Chinese college students. The study focuses on a specific region in China: Guangdong Province.
2. The article lacks discussion on what social support looks like and what it means. It is not only unclear what the authors mean by social support, and more significantly who provides it, but the role of a mediator is almost undiscussed. Social support is an act provided by certain entities or organizations or individuals. Who are these mediators offline and perhaps also online? And how does social support work? These questions are unanswered in the manuscript.
3. There's a significant flaw in the study, conveniently assuming that adding a mediating variable means the variable is in fact playing a mediative role. This is a big assumption to make. The manuscript is currently missing what this means on the ground. First missing the definition of what a mediative role is, who is mediating what, and how. And how might the survey questions on social support help characterize this and how these results connect with the narrative. Simply adding a variable in between and claiming that this has a mediative impact and because there is a statistically significant result seems not only incomplete but over-claiming based on the results.
Reviewer 4 Report
The paper was very interesting and a novel way to think about addictions and the relationship to stress. I think the manuscript would benefit with a very thorough editing process. There were several comments that would need further support. for example, lines 115 -117, where the authors say that women have an innate preference for shopping. One reference is included but this finding runs counter to their results; the authors should explain this statement more.
I especially appreciate the idea of enhanced tendencies for online shopping addictions if someone does not have strong social supports. This suggests implications for policy and practice to reduce addictions overall.
The bigger opportunity to improve the manuscript lies in the discussion and the conclusion. The results were quite detailed and there are significant implications especially for future research. The discussion section included some key points including the lack of differences based on gender.
Also the statement is made that social support is regarded as a protective factor. As well, the transition of students from secondary to post-secondary education has many challenges. These are important findings and have implications for future study. Potentially a separate section on next steps for research would be really helpful.
Highly interesting article !
